# The Effect of Enzymatic Fermentation on the Chemical Composition and Contents of Antinutrients in Rapeseed Meal

Małgorzata Kasprowicz-Potocka [ID], Anita Zaworska-Zakrzewska *[ID], Dagmara Łodyga and Damian Józefiak

Department of Animal Nutrition, Faculty of Veterinary Medicine and Animal Science, Poznan University of Life Sciences, Wolynska 33, 60-637 Poznan, Poland; malgorzata.potocka@up.poznan.pl (M.K.-P.); dagmara.lodyga@up.poznan.pl (D.Ł.); damian.jozefiak@up.poznan.pl (D.J.)
* Correspondence: anita.zaworska-zakrzewska@up.poznan.pl

**Abstract:** Enzymatic solid-state fermentation can improve the nutritional quality of feed materials. The current study aimed to determine the effect of the solid-state fermentation of rapeseed meal (RSM) with carbohydrase/s and phytases in various combinations. RSM was fermented for 24 h at 25 °C with eight commercial preparations and mixtures thereof to prepare eleven products (PHYL—liquid-6-phytase; RON NP—6-phytase; RON HI—6-phytase; RON R—α-amylase; RON WX—β-xylanase; RON VP—β-glucanase; RON A—α-amylase, β-glucanase; RON M—xylanase, β-glucanase; RON NP+M; RON NP+A; RON NP+M+R). Afterward, the enzymes were deactivated at 70 °C within 15 min, and the biomass was dried for 24 h at 55 °C. Carbohydrase and/or phytase additives did not positively affect crude or true protein content or reduce crude fiber ($p > 0.05$). Among the products after fermentation, a significant reduction in the content of the raffinose family oligosaccharides, glucosinolates, and phytate was found. In the presence of phytase, the phytate reduction was more significant ($p < 0.01$) than that in the presence of carbohydrases only. The addition of carbohydrases together with phytases did not improve the results in comparison with phytases alone ($p > 0.05$). The most valuable effect was found for liquid-6-phytase (PHYL).

**Keywords:** rapeseed meal; solid-state fermentation; enzymes; antinutrients

## 1. Introduction

Rapeseed is one of the top five oilseed crops in the world and is thus of great importance to the world's agricultural industry. The global production of rapeseed oil reached 31.8 million metric tons in 2022/2023 [1], which led to 41 million tons of solid waste (such as oil press meals or press cakes) [2]. This material is used mainly as livestock feed and as an alternative source for producing enzymes, antimicrobial agents, and bioactive compounds [3].

Several rapeseed chemicals, such as phytic acids, glucosinolates, oligosaccharides, and phenolic compounds, are considered significant disadvantages of rapeseed-derived proteins for food and feed applications [4]. Many studies have shown that the high dietary inclusion of raffinose family sugars in feeds containing high levels of crude fiber and other antinutrients (ANFs) could lead to decreased feed intake and body weight gain in nonruminants [5–7]. The most popular method for deactivating and eliminating antinutrients such as glucosinolates and myrosinase is solvent extraction with hexane, but this method negatively affects the content and availability of amino acids; however, pressing only seeds does not reduce ANFs in rapeseed [8].

Solid-state fermentation (SSF) using microorganisms (bacteria, yeast, and fungi) is recognized as an inexpensive and effective method of reducing ANFs [5–7,9]. There are also many other advantages of SSF, such as the absence of a liquid phase and low water content in the product after fermentation [10]. This means that strong drying does not occur after fermentation, which is essential because rapeseed protein is easily denatured during

high-temperature and high-pressure sterilization treatment, decreasing its conversion efficiency [11]. Fermented rapeseed products are also more easily digestible and contain peptides and vitamins [12]. Moreover, fermentation significantly increases the activity of some natural enzymes, such as proteolytic enzymes, which can effectively hydrolyze rapeseed proteins into low-molecular-weight peptides and degrade toxic substances in rapeseed meal (RSM) [12,13]. On the other hand, it is also known that the glucosinolate itself is not pungent, but when it encounters the enzyme myrosinase, an aglycone is formed, which then usually rearranges into a pungent and corrosive isothiocyanate. Isothiocyanates can also be formed from glucosinolates by steam distillation and are also called mustard oils [4,5]. Moreover, during the first hours of fermentation, a slow pH reduction and Enterobacteriaceae proliferation were observed. However, over longer durations, the quantity of lactic acid bacteria increases, and the number of pathogenic bacteria decreases. Fermentation through the significant elimination of harmful bacteria and reduction in the level of glucosinolates and their resulting products improve nutritional safety, and the synthesis of short-chain fatty acids (VFA) increases the palatability and dietary value of feed [7,9,10]. Currently, RSM or rapeseed protein is mainly hydrolyzed by different enzymes to produce peptides with better protein functional properties and antioxidant and antitumor activity [4,10,12].

Enzymatic deamidation using commercial products and fermentation can significantly improve the nutritional quality of rapeseed products [9,14,15]. Microbial enzymes are commonly used as feed additives in poultry and pig feed to reduce the negative impacts of certain ANFs [16]. Adding microbial enzymes to fermentation makes the process more targeted and specific for certain ANFs or nutrients. Moreover, this approach prolongs the amount of time that enzymes are added to substrates [17]. The study of the impact of different exogenous enzyme preparations containing α-amylase, β-xylanase or β-glucanase, and/or phytase showed their positive effect on the chemical composition of the product obtained through the SSF of rapeseed cakes (RSC) and, ultimately, on the growth performance of broiler chickens [9]. These fermentation techniques tend to increase protein digestibility in fermented materials, indicating some "predigestion" of proteins, which may be caused by the activation of proteases and other enzymes produced by microorganisms during the process [15].

Moreover, fermentation leads to phytate degradation due to the activation of intrinsic phytases. One more limiting factor for the inclusion of rapeseed products in poultry and pig diets is the high level of fiber and lignin, mainly due to the high hull content, which reduces protein digestibility. The cell wall polymers abundant in rapeseed products are pectins, xylans, xyloglucans, and arabinoxylans. The disruption of cell wall polysaccharides can increase protein and energy digestibility. Adding pectinolytic enzymes and xylanase to RSM has been shown to improve the digestion of carbohydrates both in vitro and in vivo [14]. It is also commonly known that microorganisms cannot directly destroy nutrients such as fiber in rapeseed products due to their particular physical and chemical structure [9,18]. These nutrients could be accessible to microorganisms after the material has been enzymatically hydrolyzed. However, little work has focused on reducing ANFs and improving the nutritional value of RSM through the use of an approach that combines enzymatic hydrolysis and natural fermentation [15]. This work supplements our previous publication, which examined rapeseed cakes [9]. Because of the different compositions of RSM and RSC, especially in terms of protein, fat, and antinutrient levels, we expect different results from our experiments. The present study aimed to determine the effect of the solid-state fermentation of RSM with different exogenous enzymes, such as α-amylase, β-xylanase, β-glucanase, and 6-phytase, in various combinations on the chemical composition of the obtained products.

## 2. Materials and Methods

### 2.1. Fermentation Process

The RSM was purchased from a commercial manufacturing plant ("Bielmar", Bielsko-Biała, Poland). The RSM was ground (22 mm sieve), 100 g was put into glass fermenters in three replicates, and the material was thoroughly mixed with water at a mass ratio of 1:2 in plastic containers. The substrates were inoculated with enzymes (0.1% on a meal and water weight basis) and mixed. Fermentation was performed using eight commercial mono- and poly-enzyme formulations in powdered form and their mixtures to achieve 11 different combinations, as presented in Table 1. All the enzymes were obtained from DSM Nutritional Products, Mszczonów, Poland, except for OptiPhos 500 (PHYL), which was obtained from Huvepharma N.V., Antwerpen, Belgium. Fermentation was carried out for 24 h at 25 °C under anaerobic conditions. Afterward, the enzymes were deactivated at 70 °C within 15 min, while the fermented biomass was dried for 24 h at 55 °C.

**Table 1.** Scheme of the experiments.

| Name of Fermented Product | Enzyme | Name/Activity (Unit) |
|---|---|---|
| PHYL | 6-phytase | Liquid, OptiPhos 500, 500 FTU/kg |
| RON NP | 6-phytase | RONOZYME NP; 1.5 mln FYT/ kg |
| RON HI | 6-phytase | RONOZYME HiPhos; 1.0 mln FYT/kg |
| RON R | α-amylase | RONOZYME RUMISTAR, 300,000 KNU/kg |
| RON WX | endo 1,4-β-xylanase | RONOZYME WX, 200,000 FXU/kg |
| RON VP | endo-1,3(4)-β-glucanase | RONOZYME VP, 10,000 FBG/kg |
| RON A | α-amylase, endo-1,3(4)-β-glucanase | RONOZYME A, 40,000 kNU/kg, 70,000 FBG/kg |
| RON M | endo-1,4-β-xylanase; endo-1,3(4)-β-glucanase; endo-1,4-β-glucanase | RONOZYME® MultiGrain |
| RON NP+M | endo-1,4-β-glucanase, endo-1,2(4)-β-glucanase, endo- 1,2-β-xylanase, 6-phytase | RONOZYME NP and RONOZYME® MultiGrain |
| RON NP+A | α-amylase, endo-1,3(4)-β-glucanase, 6-phytase | RONOZYME NP and RONOZYME A |
| RON NP+M+R | α-amylase, endo-1,4-β-glucanase, endo-1,2(4)-β-glucanase, endo- 1,2-β-xylanase, 6-phytase | RONOZYME NP and RONOZYME® MultiGrain and RONOZYME RUMISTAR. |

### 2.2. Chemical Analysis

All the samples were ground to pass through a 0.5 mm sieve before analysis. The raw RSM (raw material) and products after fermentation were analyzed (n = 6) for dry matter (DM), crude protein (CP), crude fiber (CF), and phosphorous (P) content according to the AOAC [19]. True protein (TP) was analyzed as described by Hsu et al. [20]. Phytate-P (Phyt-P) was determined using Haug and Lantzsch's spectrometry method [21] with an acidic iron-III solution (Spectrophotometer Marcel Media, Poland). The GC method was used to analyze the composition and contents of soluble carbohydrates, as described by Lahuta et al. [22]. Carbohydrates were quantified using standards (sugars, polyols, cyclitols, oligosaccharides, and galactinol) purchased from Sigma-Aldrich (St. Louis, MO, USA). The glucosinolates in the RSM were determined by gas-liquid chromatography of trimethylsilyl derivatives of desulphated glucosinolates, as described by Raney and McGregor [23].

### 2.3. Statistical Analysis

Statistical calculations were performed using the SAS ver. 5.0. software package (SAS Institute, Inc., Cary, NC, USA). One-way ANOVA was used for comparisons of all the materials, for which *p* values were <0.01 for highly statistically significant parameters and are marked with letters. The significance of differences between the groups was calculated using Duncan's test to compare all the groups.

### 3. Results

The basic composition and phosphorus content of the fermented material compared with those of the unprocessed RSM are presented in Table 2. The dry matter content in all the fermented materials was higher ($p < 0.0001$) than that in the raw RSM. The crude protein content in the fermented products varied significantly from 37.28 to 39.40% in DM, and in the products PHYL, RON VP, RON A, and RON NP+M+R were considerably lower than those in the raw RSM. The true protein content was significantly lower ($p < 0.01$) in all the products, except for RON NP+A, RON M, and RON HI, than in the raw RSM. The crude fiber content was the lowest in RON M ($p < 0.0001$) and was also lower than that in raw RSM ($p > 0.05$). Among the fermented products, the contents in PHYL, RON R, RON WX, RON VP, RON A, and RON NP+M, CF were significantly higher than those in unprocessed RSM ($p < 0.0001$). Phosphorus levels varied among the materials and ranged in DM from 1.20% in the RON VP to 1.36% in the RON NP+M, whereas they ranged from 1.31% in the DM of the raw material. Significantly lower *p* values were found for the RON VP, RON A, and RON NP+A products. Among the fermented products, the content of phyt-P was significantly lower than that of RSM ($p < 0.0001$). In the case of the phytases used, the reduction in phytate content was significant and ranged from 72 to 85% (for RON NP+M, RON NP+A, RON NP+M+R, PHYL, RON NP, and RON HI), and the most effective reduction was found for PHYL ($p < 0.0001$). In the case of products fermented without phytase, the decrease was lower and ranged from 25 to 33%. As a result, the ratio of phyt-P/total P was the lowest in the case of PHYL and RON NP, and did not exceed 10; higher in RON HI and RON NP+M, RON NP+A, and RON NP+M+R, and did not exceed 20; and the highest in the remaining products, where it was approximately 40, in comparison with the raw RSM, where the ratio achieved was 52.

The variability of sugars before and after fermentation is shown in Table 3. The total carbohydrate content was significantly lower in all the fermented products by approximately 57 to 70% compared to that in the nonfermented RSM. The highest reduction was found in PHYL, followed by RON NP+M, RON HI, and RON NP+R+M. Generally, compared with those in raw RSM, the contents of sugars such as fructose, galactose, glucose, maltose, maltotriose, D-chiro-inositol, mannitol, and sorbitol significantly increased after fermentation, whereas the contents of saccharose, galactinol, DGG, raffinose, and stachyose (and the sum of RFO) significantly decreased ($p < 0.0001$). The most substantial reduction was found to be approximately 98%, especially in the saccharose content. Compared with that of a nonfermented meal, the RFO range was reduced by approximately 23 to 43% ($p < 0.0001$). The lowest RFO content was found in the product RON NP+M ($p < 0.0001$). No galactose, maltose, maltotriose, D-chiro-inositol, or sorbitol was found in the row RSM. In contrast, in the fermented products, these sugars appeared, except for maltose, in PHYL, RON VP, RON M, RON NP+M, and RON NP+A, and RON NP+R+M, and sorbitol appeared in RON NP, RON HI, and RON M. The composition of glucosinolates in the nonfermented and fermented RSM is presented in Table 4. The content of all the analyzed substances decreased significantly after fermentation ($p < 0.0001$). The total glucosinolate content was approximately 90% lower in comparison with that in RSM. Additionally, significant differences among the fermented products were found, with the lowest levels of these substances occurring in PHYL, RON NP, RON HI, and RON M. In addition, napoleiferyne and 4-OH-glucuronobrassicin were destroyed during fermentation (except for RON NP+A).

**Table 2.** The basic composition and total and phytic phosphorus contents in the raw and fermented rapeseed meal (% in DM).

| Item | Raw RSM | Fermented Products | | | | | | | | | | | *p*-Value | SEM |
|---|---|---|---|---|---|---|---|---|---|---|---|---|---|---|
| | | PHYL | RON NP | RON HI | RON R | RON WX | RON VP | RON A | RON M | RON NP+M | RON NP+A | RON NP+M+R | | |
| DM | 88.86 D | 93.67 AB | 93.46 AB | 93.25 BC | 93.43 ABC | 93.02 C | 93.24 BC | 93.72 A | 93.30 ABC | 93.37 ABC | 93.35 ABC | 93.34 ABC | <0.0001 | 0.12 |
| CP | 39.39 A | 38.04 BCD | 38.46 ABC | 38.92 AB | 38.75 ABC | 38.79 ABC | 37.89 BCD | 37.28 D | 39.29 A | 38.34 ABC | 39.40 A | 37.77 CD | <0.0001 | 1.15 |
| TP | 34.17 AB | 32.70 C | 32.97 C | 33.18 BC | 32.94 C | 32.89 C | 32.32 CD | 31.53 D | 33.37 ABC | 32.81 C | 34.33 A | 32.40 CD | 0.0003 | 1.29 |
| CF | 15.52 EF | 16.71 A | 15.95 BCDE | 15.85 CDE | 16.33 ABC | 16.09 BCD | 16.39 ABC | 16.21 BCD | 15.17 F | 16.41 AB | 15.75 DE | 15.82 DE | <0.0001 | 0.64 |
| P | 1.31 ABC | 1.33 AB | 1.32 AB | 1.31 ABC | 1.29 BC | 1.29 BC | 1.20 D | 1.26 C | 1.32 AB | 1.36 A | 1.25 C | 1.30 ABC | <0.0001 | 0.07 |
| Phyt-P | 0.68 A | 0.10 I | 0.12 H | 0.16 G | 0.49 C | 0.51 B | 0.45 E | 0.47 D | 0.49 D | 0.19 F | 0.19 F | 0.18 FG | <0.0001 | 0.02 |
| Phyt-P/P | 52.0 A | 7.5 F | 9.3 F | 12.5 E | 37.9 BC | 39.5 B | 37.5 C | 37.3 C | 36.7 C | 13.8 DE | 15.4 D | 13.6 DE | <0.0001 | 1.70 |

RSM—rapeseed meal; DM—dry matter; CP—crude protein; TP—true protein; CF—crude fiber; Phyt-P—phytate phosphorous; P—phosphorus; SEM—standard error of mean; PHYL—liquid-exogenous-6-phytase; RON NP—6-phytase; RON HI—6-phytase; RON R—α-amylase; RON WX—endo 1,4-β-xylanase; RON VP—endo-1,3(4)β-glucanase; RON A—α-amylase, endo-1,3(4)-β-glucanase; RON M—endo-1,4-β-xylanase, endo 1,3(4)-β-glucanase, endo-1,4-β-glucanase; RON NP+M—endo-1,4-β-glucanase, endo-1,2(4)-β-glucanase, endo-1,2 β-xylanase, 6-phytase; RON NP+A—α-amylase, endo-1,3(4)-β-glucanase, 6-phytase; RON NP+M+R—α-amylase, endo-1,4-β-glucanase, endo-1,2(4)-β-glucanase, endo-1,2-β-xylanase, 6-phytase. [A, B, C, D, E, F, G, H, and I]—values in the same rows with different letters differ significantly at *p* < 0.001.

**Table 3.** The composition and content of soluble carbohydrates, including RFO (mg/g in DM), in the raw and fermented rapeseed meal.

| Item | Raw RSM | Fermented Products | | | | | | | | | | | *p*-Value | SEM |
|---|---|---|---|---|---|---|---|---|---|---|---|---|---|---|
| | | PHYL | RON NP | RON HI | RON R | RON WX | RON VP | RON A | RON M | RON NP+M | RON NP+A | RON NP+R+M | | |
| Fructose | 1.01 F | 4.38 E | 8.08 D | 5.52 E | 15.11 A | 13.59 A | 9.94 CD | 12.85 AB | 11.02 BC | 10.04 CD | 9.79 CD | 9.39 CD | <0.0001 | 0.48 |
| Galactose | 0.00 G | 0.18 F | 0.25 E | 0.19 F | 0.34 AB | 0.32 BC | 0.37 A | 0.31 BCD | 0.28 DE | 0.27 E | 0.25 E | 0.23 E | <0.0001 | 0.01 |
| Glucose | 1.60 DE | 1.39 E | 2.26 C | 1.61 E | 3.27 A | 3.02 AB | 2.39 C | 3.00 AB | 2.70 ABC | 2.56 BC | 2.35 C | 2.10 CD | <0.0001 | 0.09 |
| Saccharose | 75.79 A | 0.49 B | 0.56 B | 0.61 B | 0.64 B | 0.79 B | 0.88 B | 0.90 B | 1.21 B | 0.58 B | 0.59 B | 0.82 B | <0.0001 | 1.86 |
| Maltose | 0.00 C | 0.00 C | 0.25 A | 0.14 B | 0.31 A | 0.14 B | 0.00 C | 0.28 A | 0.00 C | 0.00 C | 0.00 C | 0.00 C | <0.0001 | 0.02 |
| Maltotriose | 0.00 E | 0.20 D | 0.40 BC | 0.42 BC | 0.55 A | 0.42 BC | 0.34 C | 0.41 BC | 0.44 BC | 0.38 BC | 0.37 BC | 0.48 AB | <0.0001 | 0.02 |

**Table 3.** *Cont.*

| Item | Raw RSM | Fermented Products | | | | | | | | | | | p-Value | SEM |
|---|---|---|---|---|---|---|---|---|---|---|---|---|---|---|
| | | PHYL | RON NP | RON HI | RON R | RON WX | RON VP | RON A | RON M | RON NP+M | RON NP+A | RON NP+R+M | | |
| D-chiro-inositol | 0.00 F | 0.15 CDE | 0.32 B | 0.23 BC | 0.17 CD | 0.21 BCD | 0.18 CD | 0.27 AB | 0.34 A | 0.08 E | 0.12 DE | 0.15 CDE | <0.0001 | 0.01 |
| Mio-inositol | 0.70 C | 2.69 A | 0.44 D | 1.53 B | 0.37 D | 0.36 D | 0.41 D | 0.37 D | 0.40 D | 0.46 D | 0.36 D | 0.36 D | <0.0001 | 0.09 |
| Mannitol | 0.30 C | 2.02 B | 3.75 A | 4.59 A | 1.09 BC | 1.20 BC | 0.68 C | 2.13 BC | 4.14 A | 0.81 BC | 1.04 BC | 0.55 BC | 0.0003 | 0.22 |
| Sorbitol | 0.00 B | 0.23 A | 0.00 B | 0.00 B | 0.14 AB | 0.17 AB | 0.30 A | 0.14 AB | 0.00 B | 0.18 AB | 0.16 AB | 0.33 A | <0.0001 | 0.02 |
| Galactinol | 2.27 A | 1.11 BC | 1.15 B | 1.11 BC | 1.20 B | 1.20 B | 1.23 B | 1.12 BC | 1.18 B | 1.19 B | 1.13 CB | 1.03 B | <0.0001 | 0.03 |
| Raffinose | 3.97 A | 2.74 DE | 3.26 BC | 2.95 CD | 3.62 AB | 3.39 BC | 2.30 E | 3.44 B | 3.60 AB | 2.55 DE | 3.29 BC | 3.22 BC | <0.0001 | 0.07 |
| Stachyose | 27.17 A | 16.35 CD | 18.73 BC | 17.47 BCD | 20.26 B | 20.14 B | 17.37 BCD | 19.33 B | 19.99 B | 15.12 D | 18.98 BC | 18.43 BC | <0.0001 | 0.37 |
| DGG | 0.74 A | 0.59 B | 0.41 CD | 0.48 CD | 0.44 CDE | 0.45 CDE | 0.43 CDE | 0.38 E | 0.43 BCD | 0.45 CDE | 0.50 C | 0.46 CD | <0.0001 | 0.01 |
| 1-Kestose | 0.61 B | 0.00 C | 0.00 C | 0.00 C | 0.00 C | 0.00 C | 2.59 A | 0.00 C | 0.00 C | 0.00 C | 0.00 C | 0.00 C | <0.0001 | 0.09 |
| Total carbohydrate | 114.43 A | 33.92 E | 41.21 CD | 38.46 DE | 48.61 B | 46.54 B | 40.97 CD | 46.03 BC | 46.96 B | 36.89 DE | 40.06 D | 38.58 DE | <0.0001 | 1.93 |
| Only RFO | 31.14 A | 19.08 DE | 21.99 BCED | 20.42 CDE | 23.88 B | 23.54 BC | 19.66 CDE | 22.78 BCD | 23.59 BC | 17.68 F | 22.27 BCDE | 21.64 BCDE | <0.0001 | 0.43 |

RSM—rapeseed meal; DGG—di-galaktozylo glycerol; RFO—raffinose family oligosaccharides; SEM—standard error of mean; PHYL—liquid-exogenous-6-phytase; RON NP—6-phytase; RON HI—6-phytase; RON R—α-amylase; RON WX—endo 1,4-β-xylanase; RON VP—endo-1,3(4)β-glucanase; RON A—α-amylase, endo-1,3(4)-β-glucanase; RON M—endo-1,4-β-xylanase, endo 1,3(4)-β-glucanase, endo-1,4-β-glucanase; RON NP+M—endo-1,4-β-glucanase, endo-1,2(4)-β-glucanase, endo-1,2 β-xylanase, 6-phytase; RON NP+A—α-amylase, endo-1,3(4)-β-glucanase, 6-phytase; RON NP+M+R—α-amylase, endo-1,4-β-glucanase, endo-1,2(4)-β-glucanase, endo-1,2-β-xylanase, 6-phytase. [A, B, C, D, E, F, G]—values in the same rows with different letters differ significantly at *p* < 0.001.

**Table 4.** The composition and content of glucosinolates (μmol/g in DM) in raw and fermented rapeseed meal.

| Item | Raw RSM | Fermented Products | | | | | | | | | | | *p*-Value | SEM |
|---|---|---|---|---|---|---|---|---|---|---|---|---|---|---|
| | | PHYL | RON NP | RON HI | RON R | RON WX | RON VP | RON A | RON M | RON NP+M | RON NP+A | RON NP+M+R | | |
| Gluconapine | 2.44 A | 0.32 C | 0.32 D | 0.32 D | 0.43 B | 0.32 D | 0.32 D | 0.37 C | 0.32 D | 0.32 B | 0.43 B | 0.37 C | <0.0001 | 0.05 |
| Glucobrassicanapine | 0.79 A | 0.11 B | 0.11 B | 0.11 B | 0.11 B | 0.11 B | 0.11 B | 0.11 B | 0.11 B | 0.11 B | 0.11 B | 0.11 B | <0.0001 | 0.02 |
| Progoitrin | 5.96 A | 0.37 BC | 0.32 C | 0.34 C | 0.43 B | 0.43 B | 0.38 BC | 0.43 B | 0.38 C | 0.37 BC | 0.43 B | 0.43 B | <0.0001 | 0.14 |
| Napoleiferyne | 0.34 A | 0.00 B | 0.00 B | 0.00 B | 0.00 B | 0.00 B | 0.00 B | 0.00 B | 0.00 B | 0.00 B | 0.00 B | 0.00 B | <0.0001 | 0.01 |
| 4-OH-glucobrassisine | 0.68 A | 0.00 C | 0.00 C | 0.00 C | 0.00 C | 0.00 C | 0.00 C | 0.00 C | 0.00 C | 0.00 C | 0.05 B | 0.00 C | <0.0001 | 0.02 |
| Total glucosinolate | 10.20 A | 0.80 DE | 0.75 E | 0.77 DE | 1.02 B | 0.86 CD | 0.86 CB | 0.91 C | 0.80 DE | 0.91 C | 1.07 B | 1.02 B | <0.0001 | 0.23 |
| Total glucosinolate alkene | 9.49 A | 0.80 D | 0.75 D | 0.75 D | 0.96 B | 0.86 C | 0.86 C | 0.91 BC | 0.75 D | 0.86 C | 0.96 B | 0.91 BC | <0.0001 | 0.21 |

RSM—rapeseed meal; SEM—standard error of mean; PHYL—liquid-exogenous-6-phytase; RON NP—6-phytase; RON HI—6-phytase; RON R—α-amylase; RON WX—endo 1,4-β-xylanase; RON VP—endo-1,3(4)β-glucanase; RON A—α-amylase, endo-1,3(4)-β-glucanase; RON M—endo-1,4-β-xylanase, endo 1,3(4)-β-glucanase, endo-1,4-β-glucanase; RON NP+M—endo-1,4-β-glucanase, endo-1,2(4)-β-glucanase, endo-1,2 β-xylanase, 6-phytase; RON NP+A—α-amylase, endo-1,3(4)-β-glucanase, 6-phytase; RON NP+M+R—α-amylase, endo-1,4-β-glucanase, endo-1,2(4)-β-glucanase, endo-1,2-β-xylanase, 6-phytase. A, B, C, D, and E—values in the same row with different letters differ significantly at *p* < 0.001.

## 4. Discussion

In this study, we combined the natural fermentation and hydrolysis of biomass using external enzyme preparation. Solid-state fermentation can effectively improve feed quality by activating microorganisms and native enzymes present in the fermented biomass. However, some substances can be destroyed only by enzymatic hydrolysis [13,15]. In the present study, all the obtained fermented products were characterized by a lower crude and true protein content. A significant reduction in crude protein was found in products fermented with phytases (PHYL, RON VP), α-amylase and β-glucanase (RON A), α-amylase, endo-1,4-β-glucanase, endo-1,2(4)-β-glucanase, endo-1,2-β-xylanase, and 6-phytase (RON NP+M+R) in comparison with those in raw RSM. The true protein content was also significantly lower for almost all the products, except for phytase (RON HI), endo-1,4-β-xylanase, endo-1,3(4)-β-glucanase, endo-1,4-β-glucanase (RON M), α-amylase, endo-1,3(4)-β- glucanase, and 6-phytase (RON NP+A). The enzymes used differed, so it was difficult to determine the correlation between the types of enzymes used and the protein content in the fermented products. In our former research with RSC [9], all the fermented products were characterized by increased crude and true protein levels, and in both experiments, the enzymes and fermentation conditions were the same. However, the crude protein content was higher in unprocessed RSM (39%) than in unprocessed RSC (32%), which can also determine the changes. Chiang et al. [24] considered these changes to be due to differences in dry matter content, but we recalculated the nutrient content on a DM basis. Soluble protein reduction could be caused by microorganisms using available nitrogen for differentiation and growth, as observed in other works [25]. SSF also promotes protease production, which can promote protein decomposition and the degradation of the lignocellulosic matrix, reducing lignocellulosic protein bonds and indirectly increasing protein digestibility [26]. Ashayerizadeh et al. [6] reported that increased crude protein content was associated with decreased nonstructural and total carbohydrate levels in feed. In the present study, we expected a reduction in structural carbohydrates when enzymes were added, but no reduction in crude fiber was found in these fermented products. Moreover, for almost all the materials, the fiber content increased (significantly or nonsignificantly) in DM. Conversely, the soluble carbohydrate content decreased from 11.4% in the raw RSM to 3.3–4.8% in the fermented products. Interestingly, after fermentation, the lowest total carbohydrate content was found in materials fermented with phytases. It seems that soluble carbohydrate reduction was likely a result of the natural fermentation of RSM because it is easily destroyed by native microflora, such as bacteria and yeast. Allzyme SSF with seven enzyme activities (amylase, cellulase, phytase, xylanase, beta glucanase, pectinase, and protease) added at a rate of 300 g/ton of feed, also increased the NSP content in sweet potato vine meal [27]. Our previous results [9] also revealed a significant reduction in the abundance of saccharose, galactinol, raffinose, and stachyose, and the intensive changes in carbohydrate levels (also found in this research) were probably caused by the activity of native microorganisms that use non-starch polysaccharides (NSP) and simple sugars to produce their own biomass. Like in earlier research [9], in the current study, glucose and fructose levels were significantly higher in products with carbohydrases. Additionally, Boorojeni et al. [5] found that both natural and enzymatic fermentation reduced insoluble and total NSP levels in the RSC. The fermentation of RSC with enzymes resulted in a slight increase in the soluble NSP concentration. However, the current study did not achieve effective fermentation because of the lack of positive changes in protein content, which showed that the bacterial mass did not increase well. This difference was probably a result of the low water content in the substrate, which limits enzyme activity during fermentation; moreover, because of the lower fat content, the RSM was characterized by greater hygrophilicity. Moreover, rapeseed contains mainly NSPs, such as rhamnose, arabinose, xylose, mannose, and galactose [5]; therefore, the addition of amylase and β-glucanase to fermentation products may not be effective in the case of RSM. Pustjens et al. [28] also found that RSM contains mainly glucose, arabinose, and uranyl groups and assumed that the cell wall polysaccharide matrix of RSM is strongly associated with

each of them. Therefore, it was proven that fiber fractions are generally more resistant to fermentation than are other fractions, and 24 h is too short for decomposing plant cell walls. Nevertheless, the enzymes used probably also did not contain the proper substrate for the action [14]. Jakobsen et al. [29], using phytase, xylanase, xylanase + β-glucanase, phytase + xylanase; phytase + xylanase + β-glucanase, a β-glucanase + xylanase + pectinase and xylanase mixture, cellulase, and cellulase + xylanase, found that only the mixture of β-glucanase + xylanase + pectinase decreased the NSP content by approximately 31% during 48 h of fermentation. When other enzymes and enzyme mixtures were used, the NSP content increased.

The main ANFs in the RSM were raffinose family oligosaccharides, glucosinolates, and phytate. Some studies have shown that bacteria and yeasts can reduce the content of oligosaccharides (by 73%), glucosinolates (by 97.3%), and phytate (by 67%) in RSM during fermentation [14,30]. In the actual study, compared with that in nonfermented meals, the RFO content in all the products was reduced by approximately 23 to 43%, whereas the observed reduction in RSC ranged from 45 to 60% [9]. However, the content of RFOs in the RSM was higher than that in the RSC (31.4 vs. 22.9 mg/g DM). The difference between products fermented with different enzymes is unclear, which probably means that the reduction in RFO was connected mainly with natural microbial activity in the fermented material [7]. Jakobsen et al. [29] reported that all the treatments used increased numbers of lactic acid bacteria, with concomitant increases in lactic acid and acetic acid and a reduction in pH. Unfortunately, the current study did not use a control sample fermented without an enzyme for comparison. However, RFOs are known to be destroyed primarily by yeast and bacteria to form simple sugars [31,32]. In rapeseed products, Lücke et al. [33] found that *Rhizopus oligosporus* degraded polyphenols, glucosinolates, and some polysaccharides. Generally, the rapeseed varieties cultivated in Poland contain very low glucosinolate levels—below 15 μmol/g of seeds [34]; in the present study, the glucosinolate concentration was 10.2 μmol/g; and in previous studies, the glucosinolate concentration was 16.8 μmol/g of material [9]. In the current study, compared with RSM, total glucosinolates were degraded during fermentation in all the products by approximately 90%. These same results were also noted in our earlier research [9]. We think this difference could result from the presence of native microorganisms and natural myrosinase in the RSM. Myrosinase spontaneously degrades the S-glycosidic bond in glucosinolates [35] and may be activated during fermentation, especially at high temperatures. Myrosinase cleaves off the glucose group from a glucosinolate in water, which a higher glucose level in the fermented products could partially demonstrate. Many authors have shown that glucosinolate degradation depends on enzyme type and concentration, temperature, pH, and reaction time [36]. Chiang et al. [24] found that 90% of glucosinolates were reduced after 30 days of fermentation. Therefore, in the current study, the degradation of glucosinolates was satisfactory. Additionally, napoleiferyne and 4-OH-glucobrassicin were destroyed during fermentation. There is no information in the literature about napoleiferine. However, 4-OH-glucobrassicin is a derivative of glucobrassicin found in most *Brassicae* plants. Metabolites of glucobrassicin act in cell cycle regulation in cancer cells through nuclear factor-κB (NF-κB) signaling, caspase activation, estrogen metabolism, estrogen receptors (ERs), endoplasmic reticulum stress, and breast cancer gene expression [37]. For all the types of fermented products, an intensive reduction in phytate-P content was found. Among the products where phytases were used, phytate was reduced by approximately 72–85%, with the most effective being liquid-6-phytase. The reduction was lower for the remaining products, from 25% to 33%. In our earlier work [9], a total reduction was found in all the variants in which phytase was added. The action of phytase affected a significant pool of myo-inositol, mainly in PHYL and RON HI. Myo-inositol is essential for the normal functioning of cells so that these changes can occur [9]. Native phytase is also present in almost all plants, but its effectiveness depends on its activity, pH, and temperature. Boorojeni et al. [5] compared natural fermentation and fermentation with an enzyme mixture (phytase—RONOZYME® HiPhos and a blend of pectinases and β-glucanase—RONOZYME® VP) for 24 h at 35 °C

and reported that the enzymatic fermentation process drastically reduced the phytic acid concentration in comparison with that in untreated RSC (29.65 vs. 0.87 mg/g DM fat-free), while fermenting RSC without enzymes had no impact on the phytic acid content. The optimum activity of plant phytase occurs at 45–55 °C and a pH of approximately 5, whereas microbial phytase has optimum activity at 55–65 °C and a pH of 2.5 or 5.5 [38]. In this study, the fermentation temperature was only 25 °C, but the pH was commonly reduced to 4–5; therefore, some native enzymes may also partially reduce phytate-P. In the current research, the effectiveness of all the phytases was apparent, although the results obtained during rapeseed cake fermentation were more spectacular. In addition, the content of phytate-P was 2 times greater in the RSM than in the RSC (0.68 vs. 0.31%) [9].

In summary, the addition of carbohydrases and/or phytases did not increase crude or true protein content or reduce crude fiber. A significant reduction in the content of raffinose family oligosaccharides, glucosinolates, and phytate was observed in all the variants. In the presence of phytase (PHYL, RON NP RON HI, or mixed enzymes with phytase—RON NP+M, RON NP+A, or RON NP+M+R), the phytate reduction was higher. The addition of all the carbohydrases and phytases did not improve the results compared to phytases. This means that the conditions during solid-state fermentation did not allow for the activity of the carbohydrases used. Therefore, the most valuable effects were found for liquid-6-phytase because the lowest level of all the analyzed antinutrients was found in the RSM fermented with this enzyme. These results also correspond with our previous research, where this same enzyme was the most effective during the natural fermentation of RSC. It is also crucial that the material be dosed in liquid form, which allows the loss of enzyme activity to be avoided, resulting in a more stable product and greater flexibility in dosing. The results obtained during SSF under these conditions for RSM and RSC differed, especially regarding the lower increase in protein in the RSM products. Additionally, phytases presented different activities related to these two materials because they reduced phytate-P in total in the RSC but not in the RSM, which could be an effect of the other chemical characteristics of both materials. This means that the conditions during solid-state fermentation did not allow for the activity of the carbohydrases to be manifested. Therefore, the most valuable effects were found for liquid-6-phytase because the lowest level of all the analyzed antinutrients was found in the RSM fermented with this enzyme.

## 5. Conclusions

The direct solid-state fermentation of rapeseed meal with exogenous xylanase, β-glucanases, and/or phytase did not improve the content of nutrients, such as protein and crude fiber. Processing significantly reduced phytate, raffinose family oligosaccharide, and glucosinolate contents. In the presence of phytase, phytate-P was reduced more than it was where phytase was upset. Using carbohydrase for the solid-state fermentation of rapeseed meal was less effective than using phytases. Using carbohydrases and phytase together did not improve the nutritional value of fermented rapeseed meal compared to that of phytase alone. These findings showed that the conditions used during solid-state fermentation did not allow for the activity of the carbohydrases. The most valuable effects were found for liquid-6-phytase.

**Author Contributions:** Conceptualization, A.Z.-Z. and M.K.-P.; methodology, A.Z.-Z., M.K.-P. and D.J.; software, A.Z.-Z; validation, M.K.-P., D.J. and D.Ł. formal analysis, M.K.-P. and A.Z.-Z.; investigation, M.K.-P. and A.Z.-Z.; resources, A.Z.-Z. and D.J.; data curation, A.Z.-Z. and D.Ł.; writing—original draft preparation, M.K.-P. and A.Z.-Z.; writing—review and editing, M.K.-P. and A.Z.-Z.; visualization, M.K.-P. and D.J.; supervision, D.J.; project administration, A.Z.-Z. and D.J.; funding acquisition, A.Z.-Z. and D.J. All authors have read and agreed to the published version of the manuscript.

**Funding:** This study was supported by Programme BIOSTRATEG1/267659/NCBR/2015 "GUTFEED—INNOVATIVE NUTRITION FOR SUSTAINABLE POULTRY PRODUCTION" and a subsidy from the Ministry of Science and Higher Education of Poland, and by Poznań University of Life Sciences (Poland) through the research program "First grant", no. 1/2022.

**Institutional Review Board Statement:** Not applicable.

**Informed Consent Statement:** Not applicable.

**Data Availability Statement:** The data are available upon reasonable request to the corresponding authors.

**Conflicts of Interest:** The authors declare no conflicts of interest. The funders had no role in the design of the study; in the collection, analyses, or interpretation of the data; in the writing of the manuscript; or in the decision to publish the results.

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
