# Peer review of "The Effect of Enzymatic Fermentation on the Chemical Composition and Contents of Antinutrients in Rapeseed Meal"

_fermentation, doi:10.3390/fermentation10020107_

Round 1

Reviewer 1 Report

Comments and Suggestions for Authors

The manuscript entitled The Effect of Enzymatic Fermentation on Chemical Composition and Contents of Antinutrients in Rapeseed Meal  researched on the fermentation of rapeseed meal with different enzymes. Authors used the different enzymes, but the fermentation was all carried out at 25 C, did these enzymes showed optimal activities at 25 C? moreover, only one volume of addition was used in current study, is this addition amount is the best for enzymatic fermentation of rapeseed? The experiments are not sufficient. It is obviously enzymes will improve the quality of product, however, only limited information was shown in this manuscript. 

Comments on the Quality of English Language

language needs to be improved. 

Author Response

Dear Reviewer,

On behalf of my co-authors and myself, I would like to thank the Reviewer for the very detailed comments and suggestions to our manuscript, which greatly helped to improve our manuscript.  The detailed reply to each of the comments is presented below, including the number of lines where it leads to a change in the paper. All the changes were marked as red.

The actually version of the paper was adjusted according to those suggestions.

Sincerely,

Anita Zaworska-Zakrzewska

Reviewer 1.

  1. Authors used the different enzymes, but the fermentation was all carried out at 25 C, did these enzymes showed optimal activities at 25 C?

Our earlier small-scale pilot studies (unpublished) showed that running the process under these conditions can be very effective and there is no need to increase the temperature for this set of enzymes (process conditions). This was important for us from the point of view of the cost-effectiveness of the process.

  1. moreover, only one volume of addition was used in current study, is this addition amount is the best for enzymatic fermentation of rapeseed? The experiments are not sufficient. It is obviously enzymes will improve the quality of product, however, only limited information was shown in this manuscript.

 Thank you for your question. The first laboratory test (in the small scale)  showed that this level of enzymes was the most optimal and also not costly/expensive for a commercial process.

Additionally, as we indicated in the introduction of our article, this work supplements our previous publication, which examined other rapeseed product (rapeseed cakes) but the same enzymes and the same methodology. (Zaworska-Zakrzewska, A.; Kasprowicz-Potocka, M.; Kierończyk, B.; Józefiak, D. The Effect of Solid-State Fermentation on the Nutritive Value of Rapeseed Cakes and Performance of Broiler Chickens. Fermentation 2023, 9(5), 435. https://doi.org/10.3390/fermentation9050435 )

Because of the different compositions of rapeseed meal and rapeseed cake, especially in the content of protein, fat, and antinutrient levels, we expected different results from experiments.

  1. language needs to be improved – the manuscript was verified again by native speaker.

Reviewer 2 Report

Comments and Suggestions for Authors

This article is a interesting article on the effect of enzymatic fermentation on the RSM. The article is, well written, interesting and well developed. Nevertheless, Some things have to be improved. The results are interesting and complex, it is better to cite more references to explain fully in the discussion part, for example, for the decreased content protein. 

Author Response

Dear Reviewer,

On behalf of my co-authors and myself, I would like to thank the Reviewer for the very detailed comments and suggestions to our manuscript, which greatly helped to improve our manuscript. The detailed reply to each of the comments is presented below, including the number of lines where it leads to a change in the paper. All the changes were marked as red.

The actually version of the paper was adjusted according to those suggestions.

Sincerely,

Anita Zaworska-Zakrzewska

Reviewer 2.

The results are interesting and complex, it is better to cite more references to explain fully in the discussion part, for example, for the decreased content protein – Thank you for comment. Some new positions in discussion part were added [25, 26, 27, 37]. – References – positions marked in red [Line 371-377 and 409-411]

Reviewer 3 Report

Comments and Suggestions for Authors

This study explores the impact of solid-state fermentation on rapeseed meal (RSM) using various combinations of carbohydrases and phytases. The fermentation process involved 24 hours at 25°C with eight commercial enzyme preparations, resulting in eleven products. The addition of carbohydrases and/or phytases did not significantly affect crude and true protein or reduce crude fiber. However, all fermented products showed a notable decrease in raffinose family oligosaccharides, glucosinolate, and phytate. Phytase presence led to a more significant reduction in phytate compared to carbohydrases alone. Interestingly, combining carbohydrases with phytases did not enhance outcomes compared to phytases alone. The study highlights the most beneficial effect observed with liquid-6-phytase (PHYL).

It is a relevant study, properly presented.

To be considered:

11 RSM was fermented by 24 h at 25°C --> RSM was fermented for 24 h at 25°C

15 and dried for 24 hours at--> does this mean only the enzymes were dried?

What is the significance of reducing antinutrients in rapeseed meal (feed trials were not done)? Any disadvantages?

27 biopolymers: Which biopolymers are made from the press cake? It this already commercial, as implied?

79 0.1% on a meal weight basis -->is this not a high number for enzyme addition (costly for a commerical process)?

83 Fermentation was carried out by 24 h at 25°C --> Fermentation was carried out for 24 h at 25°C

96 was calculated using the standard internal method: Can you give details or a reference?

101 One-way ANOVA for comparison of all the materials were used--> was used

140  The napoleiferyne and 4-OH-glucobrassicin: What is the significance of these 2 compounds?

The conclusion is very short and should be expanded.

Comments on the Quality of English Language

minor editing needed

Author Response

Dear Reviewers,

On behalf of my co-authors and myself, I would like to thank the Reviewers for the very detailed comments and suggestions to our manuscript, which greatly helped to improve our manuscript.  The detailed reply to each of the comments is presented below, including the number of lines where it leads to a change in the paper. All the changes were marked as red.

The actually version of the paper was adjusted according to those suggestions.

Sincerely,

Anita Zaworska-Zakrzewska

Reviewer 3.

L.11. RSM was fermented by 24 h at 25°C --> RSM was fermented for 24 h at 25°C – it was improved;

L.15. and dried for 24 hours at--> does this mean only the enzymes were dried?  - it was improved “…and biomass was dried for 24 hours..”;

 What is the significance of reducing antinutrients in rapeseed meal (feed trials were not done)? Any disadvantages?  Free trials were not done. Some new information was added.

From the other side it is also known that the glucosinolate itself is not pungent, but when it encounters the enzyme myrosinase, the aglycone is formed which then usually rearranges into a pungent and corrosive isothiocyanate. Isothiocyanates can also be formed from glucosinolates by steam distillation and so are called mustard oils [4,5]. Moreover, during the first hours of fermentation there is observed a slow pH reduction, but also the proliferation of Enterobacteriaceae. However, in longer time the quantity of lactic acid bacteria increases and reduces pathogenic bacteria. Fermentation process by significant elimination harmful bacteria and reducing the level of glucosinolates and resulting products their breakdown improves nutritional safety, and through the synthesis of short-chain fatty acids (VFA) increases palatability and dietary value of the feed [7,9,10]. Line 42-53.

  1. 27 biopolymers: Which biopolymers are made from the press cake? It this already commercial, as implied? It was not precisive – biopolymers are made from oil. “Biopolymers” was removed.
  2. 79 0.1% on a meal weight basis -->is this not a high number for enzyme addition (costly for a commerical process)? The substrates were inoculated with enzymes (0.1% on a meal and water weight basis) and mixed - it was improved.
  3. 83 Fermentation was carried out by 24 h at 25°C --> Fermentation was carried out for 24 h at 25°C – it was improved.
  4. 96 was calculated using the standard internal method: Can you give details or a reference? It was a mistake, because this sentence was connected with unidentified pics in the method of different chemical analysis described in source [22]. It was removed.
  5. 101 One-way ANOVA for comparison of all the materials were used--> was used – it was improved

L.140 The napoleiferyne and 4-OH-glucobrassicin: What is the significance of these 2 compounds? -  it was added.

There is no information in the literature about napoleiferine. However, 4-OH-glucobrassicin it is a derivative of glucobrassicin found in most Brassicae plants. Metabolites of glucobrassicin act on cell-cycle regulation in cancer cells, with actions on nuclear factor-κB (NF-κB) signaling, caspase activation, estrogen metabolism, estrogen receptors (ERs), endoplasmic reticulum stress, and breast cancer gene expression (Šamec et al., 2018). Line 235-240.

The conclusion is very short and should be expanded.  - It was changed. Line 279-287

The direct solid-state fermentation of rapeseed meal with exogenous carbohydrases and/or phytase did not improved the content of nutrients as protein and crude fiber. Process significantly reduced phytate, raffinose oligosaccharides, and glucosinolate content. In the presence of phytase, phytate-P was reduced better than where phytase was upset. Using carbohydrase to solid-state fermentation of rapeseed meal was less effective than phytases. Using carbohydrases and phytase together did not improve the nutritional value of fermented rapeseed meal compared to phytase alone. It shows that conditions during solid-state fermentation did not allow for manifesting the activity of used carbohydrases. The most valuable effects were found for liquid-6-phytase.

Comments on the Quality of English Language minor editing needed - the manuscript was verified again by native speaker.

Round 2

Reviewer 1 Report

Comments and Suggestions for Authors

Authors mentioned this "Our earlier small-scale pilot studies (unpublished) showed that running the process under these conditions can be very effective" in their response, however, it will be better to show the results in current manuscript. Please supplement the results regarding the optimal temperature and addition of enzymes in the results session. 

Due to the difference in raw material, there will be difference in the results, but it is important to reveal and understand the mechanism for further utilization of these enzymes. 

Comments on the Quality of English Language

Minor editing of language is needed. 

Author Response

Dear Reviewer,

Thank you for taking up another review our manuscript.

Yes, we agree (and it is obvious) that due to the difference in raw material, may will be difference in the results. Our earlier small-scale pilot studies with some different feed showed that running the process under described in the manuscript conditions the most effective.During on this study we optimized parameters for several raw materials and I and my team would like to publish these results as one in another manuscript (doctoral dissertation) that's why it is not possible to add them to this work.